# SARS-CoV-2 Antibody Effectiveness Is Influenced by Non-Epitope Mutation/Binding-Induced Denaturation of the Epitope 3D Architecture

**DOI:** 10.3390/pathogens11121437

**Published:** 2022-11-29

**Authors:** Moffat M. Malisheni, Matthew Bates, Albert A. Rizvanov, Paul A. MacAry

**Affiliations:** 1Institute for Biomedical Sciences, Georgia State University, Atlanta, GA 30303, USA; 2Department of Microbiology and Immunology, Yong Loo Lin School of Medicine, National University of Singapore, Singapore 117597, Singapore; 3Immunology Programme, Yong Loo Lin School of Medicine, National University of Singapore, Singapore 117597, Singapore; 4Department of Life Sciences, Lincoln University, Lincoln LN6 7TS, UK; 5Institute of Fundamental Medicine and Biology, Kazan (Volga Region) Federal University, 420008 Kazan, Russia

**Keywords:** SARS-CoV-2 VOC, spike/RBD protein, AASC repositioning, hydrogen bond, donor/acceptor atoms

## Abstract

The public health threat from severe acute respiratory syndrome coronavirus 2 (SARS-CoV-2) continues to intensify with emerging variants of concern (VOC) aiming to render COVID-19 vaccines/infection-induced antibodies redundant. The SARS-CoV-2 spike protein is responsible for receptor binding and infection of host cells making it a legitimate antibody target. Antibodies mostly target epitopes in the receptor binding domain (RBD). Mutations occurring within epitopes influence antibody specificity and function by altering their 3D architecture. However, the mechanisms by which non-epitope mutations in the RBD influence antibody specificity and function remain a mystery. We used Protein Data Bank (PDB) deposited 3D structures for the original, Beta, Delta, BA.1, and BA.2 RBD proteins in complex with either neutralizing antibodies or Angiotensin-Converting Enzyme 2 (ACE2) to elucidate the structural and mechanistic basis for neutralizing antibody evasion driven by non-epitope amino acid substitutions in the RBD. Since the mechanism behind the extensively reported functional discrepancies between the same antibody when used individually and when used in an antibody cocktail is lacking, we explored the structural basis for this inconsistency. Finally, since SARS-CoV-2 antibodies are viral mutagens, we deciphered determinants for antibody-pressured amino acid substitutions. On the one hand, we show that non-epitope mutations in the RBD domain of SARS-CoV-2 VOC influence the formation of hydrogen bonds in the paratope-epitope interface by repositioning RBD amino-acid sidechains (AASCs). This increases the distance between complementary donor/acceptor atoms on paratope and epitope AASCs leading to weaker or the complete prevention of the formation of hydrogen bonds in the paratope-epitope interface. On the other hand, we show that SARS-CoV-2 VOC employ the same strategy to simultaneously search for complementary donor/acceptor atoms on ACE2 AASCs to form new interactions, potentially favoring increased viral transmission. Additionally, we illustrate that converting the spike protein to an RBD, a deletion mutation, also repositions epitope AASCs and that AASC interactions in the paratope-epitope interface vary when an antibody is used individually versus when utilized as a cocktail with other antibodies. Finally, we show that the process of substituting immunogenic RBD amino acids begins with the repositioning of their AASCs induced by immune/antibody pressure. We show that donor/acceptor atoms from any amino acid can determine cross-reactivity instead, provided they possess and present spatially pairing donor/acceptor atoms. By studying structural alignments for PDB deposited antibody-RBD 3D structures and relating them to published binding and neutralization profiles of the same antibodies, we demonstrate that minor structural alterations such as epitope AASC repositioning have a major impact on antibody effectiveness and, hence, should receive adequate attention given that protein structure dictates protein function.

## 1. Introduction

Protein structures are determined by amino acid sequences and are considered denatured when their biological, chemical, and physical properties are altered due to distortions in their 3D native structures [1]. Distortion of protein structures can be caused by several factors including chemical denaturants, pH, force, pressure, temperature, and mutation [2]. Mutations in severe acute respiratory syndrome coronavirus 2 (SARS-CoV-2) have produced Alpha, Beta, Gamma, Delta, and Omicron, also known as variants of concern (VOC) since they exhibit increased capabilities for immune evasion, transmissibility, and are highly pathogenic [3]. Omicron is the predominant VOC with several subvariants including the newly reported BA.4 and BA.5 poised to supplant BA.1 and BA.2 [4]. The spike protein present on the surface of SARS-CoV-2 comprises the N-terminal S1, consisting of the N-terminal domain (NTD) and a receptor-binding domain (RBD), and the C-terminal S2 domain. While the S2 domain is responsible for viral fusion leading to host cell infection, the S1 domain through the RBD attaches the SARS-CoV-2 virus to angiotensin-converting enzyme 2 (ACE2) receptors on host cells to initiate infection [5]. Hence, COVID-19 vaccines and therapeutic antibodies are designed to target the spike protein, precisely the RBD [5,6]. 

Donor/acceptor atoms on original RBD and paratope/ACE2 amino-acid sidechains (AASCs) facilitate the formation of hydrogen bonds [7]. Hydrogen bonds are essential for both stabilizing the RBD-paratope/ACE2 complex and enabling antibodies/ACE2 to function effectively [8]. Replacing native amino acids with alien amino acids could result in the disruption of hydrogen bonds and abrogation of the RBD-paratope/ACE2 interaction consequently. Epitope amino acid substitutions in the RBD promote immune evasion and viral transmission [9,10]. We hypothesized that viral escape can also occur when epitope AASCs are spatially repositioned since this may disrupt hydrogen bonds. Epitope AASC repositioning can be facilitated by non-epitope mutations. However, information on the effect of non-epitope mutation-induced RBD AASC repositioning on RBD-antibody and RBD-ACE2 function remains elusive. This will require deciphering the structural basis for the increased immune evasion and strengthened ACE2 engagement by constantly emerging SARS-CoV-2 VOC. Allosteric modulation of epitope 3D structures driven by non-epitope RBD amino acid substitutions can influence antibody function [11]. However, the mechanism remains uncharacterized. 

We examined several Protein Data Bank (PDB) deposited 3D structures of the original, Beta, Delta, Kappa (a variant of interest), BA1, and BA2 RBD proteins in complex with either antibodies or ACE2. 

## 2. Materials and Methods

### 2.1. Acquisition of 3D Structures

To explore the role of mutation in repositioning RBD AASCs, we obtained 3D structures from the Protein Data Bank (PDB) for the original, Beta, Delta, Kappa, and Omicron (BA1 and BA2) spike or RBD proteins in complex with either SARS-CoV-2 antibodies or ACE2. To assess the generalizability of findings, we obtained PDB 3D structures of viral proteins with functions analogous to the SARS-CoV-2 spike protein which included influenza antibodies in complex with variant hemagglutinin proteins and a Zika/Dengue cross-reactive antibody in complex with either the Zika or Dengue envelope glycoproteins. We also added 3D structures of SARS-CoV-2 antibodies in complex with either the spike or RBD proteins to explore the effect of spike protein truncation on the repositioning of RBD AASCs. Since immunoglobulins are also proteins, we included PDB structures for a Fab in complex with the HIV-gp120 glycoprotein and its parent IgG to explore the generalizability of truncation-induced AASC repositioning. Since induced-fit causes paratope and epitope AASCs to reposition upon binding, we explored the structural implications of induced-fit on the same antibody when used individually or when mixed with other antibodies in antibody cocktails. To undertake this task, we acquired PDB 3D structures for the same RBD in complex with SARS-CoV-2 antibodies either individually or as antibody cocktails. To explore the generalizability of induced-fit on epitope and paratope AASC repositioning, we extended the same analysis to PDB 3D structures for Ebola antibodies in complex with the same glycoprotein protein. Table 1 summarizes the characteristics of all 3D structures included in our analysis. Several PDB-deposited 3D structures were randomly selected with a focus on those authorized for emergency use by the United States Food and Drug Administration (FDA). For illustrational purposes, the selected 3D structures were narrowed down to only include those 3D structures with hydrogen bonds in their paratope-epitope interface. The similarity between various protein amino acid sequences was validated using Yet, Another Scientific Artificial Reality Application (YASARA). 

### 2.2. Analysis of 3D Structures

We used YASARA View [12] (http://www.yasara.org/, accessed on 25 October 2022) to display all PDB 3D structures included in the analysis. We analyzed the length of hydrogen bonds because it determines the strength of RBD-antibody/ACE interactions. To analyze the characteristics of hydrogen bonds and highlight the presence of mutation-induced AASC repositioning in the RBD-antibody/ACE2 interface, the original SARS-CoV-2 RBD was structurally aligned with the Beta, Delta, Kappa, BA.1, and BA.2 RBD proteins using MUSTANG [13]. The spike and RBD proteins belonging to the same SARS-CoV-2 variant were structurally aligned to demonstrate the effect of the truncation on the repositioning of epitope AASCs. To show epitope AASC repositioning due to induced-fit, we structurally aligned RBD proteins belonging to the same SARS-CoV-2 variant but in a complex with either a single antibody or a cocktail including the same antibody. Hydrogen bond length was determined by measuring the distance between donor/acceptor atoms on paratope AASCs and acceptor/donor atoms on epitope AASCs. Hydrogen bond distance is presented in Angstroms. To clarify and highlight hydrogen bond interactions between paratope and epitope donor/acceptor atoms, we mostly exposed AASCs for the structurally aligned proteins as described above. AASCs in the paratope-epitope interface were displayed using sticks rather than ribbons, cartoons, or surfaces to both simplify our analysis and prevent the obstruction of hydrogen bond interactions. The same analysis was employed for non-SARS-CoV-2 3D structures Table 1.

### 2.3. Analysis of Immune-Pressured AASCs

Prism-GraphPad was used to generate histograms depicting the repositioning of donor/acceptor atoms on epitope AASCs driven by antibody pressure. To achieve this, hydrogen bond distances determined by YASARA were converted to percentages from Angstroms using [(M − N)/M] × 100%, where M is the largest distance between the same donor/acceptor atoms on paratope AASCs for the same antibody and complementary acceptor/donor atoms on epitope AASCs for the Beta, Delta, Kappa, BA1, and BA2. Analogous to M, N is the donor/acceptor atom distance obtained between the original SARS-CoV-2 variant and the antibody. 

## 3. Results

### 3.1. Mutation-Driven AASC Repositioning Influences the Formation of Hydrogen Bonds between the Paratope and RBD Complementary Donor/Acceptor Atoms

Antibody effectiveness varies across SARS-CoV-2 VOC [14,15,16,17]. The structural basis for this discrepancy remains unknown. We show that mutations in the SARS-CoV-2 VOC RBD protein cause epitope AASCs to reorient (Figure 1A–F). This increases the distance between complementary donor and acceptor atoms. Since hydrogen bonds will not form in proteins when the distance between the donor and acceptor atoms is beyond 3.5 Angstroms (Figure 1G), the increase in distance between complementary donor and acceptor atoms might result in either the weakening or disruption of hydrogen bonds in the paratope-epitope interface. We explored the generalizability of these findings to Influenza and Zika antibodies. Mutation-induced AASC repositioning on Influenza and Zika/Dengue surface variant proteins alter their interactions with respective paratope donor/acceptor atoms (Appendix A; AF4H1K1-C) and result in the disruption of hydrogen bonds (Appendix A; AF4H1K1D, F045-092D–E, and Z021D). Interestingly, we show that substituted amino acids can maintain epitope-paratope interactions if they spatially present the same donor/acceptor atoms as the wild-type amino acid (Appendix A Z021C). 

We extended the same analysis to ACE2 in complex with SARS-CoV-2 VOC RBD proteins (Appendix A). We show that interacting AASCs in the ACE2-RBD interface varied depending on the SARS-CoV-2 VOC. For example, G502 on all VOC including the original SARS-CoV-2 but not on the Beta variant interacted with ACE2. Similar observations were made by others for BA.1, BA.1.1, BA.2, and BA.3 [10]. Based on hydrogen bond distances between K353 (ACE2)-G502 (RBD) and G496 (ACE2)-G502 (RBD), it will not be unreasonable to conclude that the original, Delta, Kappa, BA.1, and BA.2 may bind ACE2 with different strengths (Appendix A). ACE2 Q42 interacts with Q498 on the original RBD, while in the Delta variant it interacts with G446. ACE2 Q42 interacts with Q498 on the original RBD while the mutant R498 in BA.1 interacts with ACE2 D38. These seemingly minor structural variations have major functional implications [10]. 

### 3.2. Truncating the Spike Protein Repositions Epitope AASCs Which Influences the Formation of Hydrogen Bonds

Structural evidence in support of functional discrepancies between the spike and RBD proteins of the same SARS-CoV-2 variant remains elusive. We structurally aligned spike and RBD proteins belonging to the same SARS-CoV-2 variant in a complex with the same SARS-CoV-2 antibody (Figure 2A–C). Interacting paratope-epitope AASCs were exposed to highlight the effects of truncating the spike protein and how this influences the formation of hydrogen bonds. We show using four different antibodies, A19-61.1, B1.182.1, BD-368, and S309, that converting the spike protein to RBD reoriented epitope AASCs in the RBD which resulted in RBD but not spike proteins forming hydrogen bonds in the paratope-epitope interface (Figure 2D). Analogous to the spike and RBD of the same SARS-CoV-2 variant, we hypothesized that truncating immunoglobulin g to form a fragment antigen binding (Fab) should exhibit similar structural distortions since antibodies are proteins. We structurally aligned the Fab and IgG variants of the same HIV antibody, b12 (Appendix A). Paratope amino acids were flashed out to expose the structural positioning of corresponding AASCs on the Fab and its parent IgG. We show that paratope AASCs in the two antibody variants assume different spatial positions (Appendix A). IgG and Fab paratope AASCs interacted with the gp120 surface differently (Appendix A). B12 R28 and W100 were positioned away from the surface in IgG but were buried in the Fab (Appendix A). These minor structural distortions may have severe functional repercussions for truncated variants since they do not assume their native 3D conformations [18,19,20,21]. 

### 3.3. Interactions by the Same Antibody Vary Depending on Whether They Are Used Individually or as an Antibody Cocktail

Antibody-RBD binding can use either the “lock and key” or “handshake” mechanism [22,23]. The “handshake” or “induced-fit” mechanism requires paratope and epitope AASCs to spatially readjust for optimal binding to occur. Cocktail and individual antibody-bound RBD proteins (Table 1) belonging to the same SARS-CoV-2 variant RBD were structurally aligned (Figure 3A). In Figure 3B we show that R97 on the ADZ8895 antibody interacted with S477 (RBD) when used as a cocktail but interacted with T478 (RBD) when used individually. Mixing antibody BD-368 with either antibodies BD-236, BD-604, or BD-629 resulted in three distinctive hydrogen bond interactions in their paratope-RBD interfaces (Figure 3B). For example, N487 in the BD-368/BD-236 cocktail forms two hydrogen bonds with R97 when used as a cocktail but only a single hydrogen bond when utilized individually. When the combination is changed to BD-368/BD-604 the interactions with R97 are maintained but N32 is additionally engaged. Finally, all hydrogen bonds are lost when the BD-368/BD-629 cocktail is used. This underpins the significance of minor structural perturbations in influencing the formation or disruption of optimal paratope-epitope interactions and could, potentially, result in variations in functional activity. To explore the generalizability of these observations, Ebola virus glycoproteins belonging to the same strain in a complex with either a cocktail of C13C6 and C2G4 or C13C6 and C4G7 antibodies were structurally aligned. We show that the swapping of antibody pairs resulted in the disruption of hydrogen bonds (Appendix A; C13C6). We also structurally aligned RBD proteins belonging to the same SARS-CoV-2 variant in complex with either an individual ACE2 protein or a cocktail of ACE2 with S304 and S309 antibodies. We show that the amino acid D38 on ACE2 interacted with the atom NH1 of R498 on RBD when reacted individually and with the atom NH2 when used as a cocktail (Appendix A; ACE2). We show the presence of a hydrogen bond between S19 (ACE2) and N477 (RBD) in a cocktail of ACE2, S304, and S309 but not when ACE2 was used individually (Appendix A; ACE2). A possible explanation could be that antibodies in cocktails do not bind simultaneously to their antigens indicating that a higher affinity antibody might bind first followed by antibodies with low affinity. In this case, induced-fit may cause minor structural distortions of RBD AASCs which will subsequently influence how well the second antibody in the cocktail will interact with the epitope. 

### 3.4. Antibody Pressure Drives the Repositioning of Epitope AASCs and Their Subsequent Substitution

Epitope characterization normally follows antibody discovery and is essential for vaccine design. Since expensive equipment is required for X-ray 3Dlography or cryo-electron microscopy, one affordable way of characterizing epitopes is through generating escape viruses. This is done by growing viruses in the presence of antibodies (https://www.ncbi.nlm.nih.gov/pmc/articles/PMC5614395/pdf/jove-126-56067.pdf, accessed on 25 October 2022) to mimic immune pressure. Although escape SARS-CoV-2 variants are generated when original viruses are exposed to antibody pressure [24], the structural mechanism leading to epitope amino acid substitutions remains uncharacterized. Since hydrogen bonds in the paratope-epitope interface are essential for binding, we measured the distance between paratope and epitope acceptor/donor atoms on complementary AASCs. We analyzed several antibodies against the original and SARS-CoV-2 VOC. The determined distances were tabulated and used to plot histograms (Figure 4). By placing SARS-CoV-2 VOC in the order in which they first appeared, we show that antibody pressure-induced AASC repositioning precedes amino acid substitution. We looked at S371, K417, N440, D442, K444, N450, K458, S477, and Q493 RBD residues. Only S371, K417, N440, S477, and Q493 were substituted following antibody pressure. Most antibodies targeted Q493 followed by N450 and S477. We show that under immune pressure SARS-CoV-2 VOC reposition donor/acceptor atoms on epitope AASCs making them inaccessible to acceptor/donor atoms on paratope AASCs which influences the formation of hydrogen bonds in the paratope-epitope interface potentially enabling SARS-CoV-2 VOC to escape immunity. 

To decipher factors influencing the substitution of antibody-pressured epitope amino acids, we explored their contribution to stabilizing the SARS-CoV-2 RBD structure using hydrogen bonds (Appendix A). We show that SARS-CoV-2 easily replaces epitope amino acids that do not participate in stabilizing the RBD structure including S477 and Q493. K417 and E484 were substituted despite their direct role in RBD stabilization. However, their replacement AASCs do not alter the RBD structure probably because donor/acceptor atoms involved in forming stabilizing hydrogen bonds are conserved (Appendix AF). Conservation of D442, K444, N450, and K458 amino acids across all SARS-CoV-2 VOC despite being subjected to intense immune pressure might be an indication that they might possess indispensable characteristics than just conserved donor/acceptor atoms which are crucial for stabilizing the RBD protein. Epitope AASCs are merely waved around to continuously reposition their donor/acceptor atoms instead. 

The benefits of repositioning epitope AASCs are to intentionally make them collide with paratope AASCs which might prevent epitope engagement (Appendix A), inhibiting induced-fit-driven binding by hooking aromatic paratope AASCs (Appendix A), scan for proximal donor/acceptor atoms on ACE2 (Appendix A). Additional escape mechanisms include replacing shorter with longer epitope AASCs to establish contact with donor/acceptor atoms on distant ACE2 AASCs (Appendix A) or switching donor/acceptor atoms to abrogate binding with paratope donor/acceptor atoms (Appendix A). Finally, we demonstrate a trend between Q24 on ACE2 and N487 from the original SARS-CoV-2 to BA.2 in the order they emerged. Mutation-induced AASC repositioning gradually reduced the distance between Q24 (ACE2) and N487 (RBD) eventually leading to the formation of a hydrogen bond in BA.2 (Appendix A). 

## 4. Discussion

Protein binding and neutralization profiles are highly dependent on their native 3D structures which are determined by amino acid sequences. While it is known that epitope amino acid substitutions influence RBD-antibody/ACE2 interactions, the role of epitope AASC repositioning induced by non-epitope mutations in promoting immune escape and ACE2 binding simultaneously remains unexplored. We show that non-epitope mutations, including truncations, and induced-fit can reposition RBD AASCs leading to the weakening, complete disruption, or establishment of new hydrogen bonds with paratope/ACE2 AASCs. Discordant epitope-paratope/ACE2 interactions facilitated by structural variations might explain inconsistencies in the effectiveness of the same antibodies/ACE2 against SARS-CoV-2 VOC and when antibodies are used in cocktails relative to individual use [10,11,14,16,25,26]. Additionally, truncation-induced structural differences might also explain discordant effectiveness reported for Fabs relative to their parent immunoglobulins [18,19,20,27] which traditionally has been explained by the difference in the number of antigen-binding sites (antibody valency) [28,29]. However, this unproven hypothesis does not explain why the same Fab binds and neutralizes some but not other subvariants of the same virus [18,19,20,27]. Another study explored the role of antibody valency in virus neutralization by comparing F(ab’)_2_ and Fab variants to their parent immunoglobulin. They found that while the F(ab’)_2_ exhibited reduced activity, the Fab completely lost activity. These researchers repeated the same experiment but this time using a different antibody and found that the Fab did not lose its binding and neutralization activity relative to the parent immunoglobulin. Therefore, they fairly concluded that antibody valency does not explain the loss of Fab activity [30]. Furthermore, if antibody valency determined antibody binding and neutralization characteristics, then decavalent immunoglobulin m would be more effective than tetravalent immunoglobulin-A which should in turn be more effective than bivalent immunoglobulin g possessing identical variable regions. However, this is not always the case given that findings are mixed [31,32], just like with Fabs and parent immunoglobulins. Based on our structural findings, we propose that whether truncated proteins/antibodies will conserve their binding and neutralization profiles will largely depend on variable region amino acid sequences, the potential for variable region AASCs to undergo induced-fit required to optimize binding, circumvent clashes with epitope AASCs or glycans [27], how severe epitope/paratope AASCs are repositioned and whether these denatured AASCs will result in clashes or in the disruption of critical paratope-epitope interactions necessary for antibodies to properly engage their respective antigens. Interestingly, b12 Fab and its parent immunoglobulin g inhibited HIV using different mechanisms of action [19] indicating that epitope AASC repositioning might not only influence antibody potency but the mechanism of inhibition potentially. 

Vaccines are designed to induce antibodies with spatially positioned acceptor/donor atoms on paratope AASCs to specifically interact with corresponding donor/acceptor atoms on RBD AASCs. Since vaccine-induced antibodies have fixed structures, epitope amino acid substitutions or mutation-induced AASC repositioning in the RBD of VOC could influence the effectiveness of vaccine-elicited antibodies. The spike protein from the original SARS-CoV-2 is still used in current COVID-19 vaccines despite the emergence of VOC with different amino acid sequences and RBD 3D structures. Changes in RDB native 3D structures might explain the consistent reduction in COVID-19 vaccine-induced antibody effectiveness against continuously emerging VOC [17], underscoring the urgent need for designing and developing multi-variant COVID-19 vaccines. Variant-specific COVID-19 vaccines might not be ideal given the high SARS-CoV-2 mutation rate. The FDA recently authorized the use of updated bivalent COVID-19 vaccines containing spike proteins from the original and BA.4/5 SARS-CoV-2 variants [33], consistent with our reporting and proposals. 

Waning immunity is believed to cause breakthrough infections while booster COVID-19 vaccine doses have been proposed as one way of increasing the level of waning neutralizing antibodies [34,35,36]. However, the fact that the same serum from COVID-19 vaccinated/infected individuals can be highly potent against the original or structurally similar SARS-CoV-2 D614G variant but less potent against predominant Omicron subvariants [11,37] challenges the “waning-immunity” concept. Vaccine/infection-induced antibodies are present in the serum but many of them have been rendered obsolete by structural alterations in RBD epitopes of Omicron subvariants. Therefore, booster doses from using first-generation COVID-19 vaccine formulations will likely increase the quantity and not the effectiveness of elicited antibodies since most antibodies will most likely have donor/acceptor atoms on paratope amino acids for which complementary acceptor/donor atoms on RBD amino acids are inexistent or no longer accessible because they are structurally repositioned in SARS-CoV-2 VOC including Omicron BA.2, BA.4, and BA.5. It is true that waning immunity or immune status modifies vaccine effectiveness, but they do not play a major role in driving SARS-CoV-2 breakthrough infections. Mutation-induced structural alterations in SARS-CoV-2 VOC are responsible for escaping vaccine/infection-induced immunity instead [34,38,39,40]. To prevent and control the rapid emergence, acquisition, and spread of SARS-CoV-2 VOC including the newly identified BA.4/5 subvariants, the focus must be to improve the quality and not increase the quantity of mismatched COVID-19 vaccine-induced antibodies. As of August 31, 2022, the FDA no longer recommends the use of monovalent (first-generation) but bivalent (containing both first-generation and BA.4/BA.5) COVID-19 vaccines as boosters [33], congruent with our observations and suggestions.

Furthermore, AASC repositioning has challenged the notion that amino acid conservation directly translates into maintained binding, and by extension neutralization activity. We have shown that spatially conserved donor/acceptor atoms directly translated into conserved binding and neutralization activity irrespective of the presenting amino acid. Hence, antibodies must be engineered to target conserved RBD donor/acceptor atoms to protect against current and future SARS-CoV-2 VOC. This should be extended to designing universal COVID-19 vaccines. We also showed that AASC repositioning must occur before immune-pressured epitope amino acids are eventually substituted and/or new contacts with ACE2 are established. Identifying immune-pressured AASCs and monitoring trends of donor/acceptor atoms on RBD and ACE2 AASCs will enable modeling of their immune-evasion and transmission capabilities which is essential for pandemic preparedness. Prospective mapping of escape mutations has been explored [41] concordant with our findings/proposals. AASC repositioning driven by induced-fit requires that various antibody combinations are assessed to come up with the most effective therapeutic antibody cocktails [22,23]. Our findings can be extended to the design of antibody-based diagnostic/detection kits. Upon binding, the capture antibody might induce structural changes in the protein which may influence optimal binding by the detection antibody. Our study has some limitations. We did not analyze all interactions key for binding such as salt bridges. However, our findings using hydrogen bonds can be extrapolated to salt bridges and potentially other interactions influenced by distances between complementary donor/acceptor atoms on paratope and RBD epitope AASCs. Interactions modeled by structural alignment should be interpreted with caution since binding can involve induced-fit. Nonetheless, findings obtained by structural alignment were corroborated by complete 3D structures. We did not include 3D structures for all Omicron subvariants as these were not deposited in the Protein Data Bank at the time of mining. A study published recently explored BA.2.12.1, BA.4, and BA.5 Omicron subvariants and demonstrated that these subvariants exhibited inconsistent neutralization profiles consistent with our findings [11]. Some comparisons were performed using structures that were resolved from different methods and, therefore, it is possible that differences observed using these structures might be artifacts. Nonetheless, our findings are supported by functional studies from the literature. 

## Figures and Tables

**Figure 1 pathogens-11-01437-f001:**
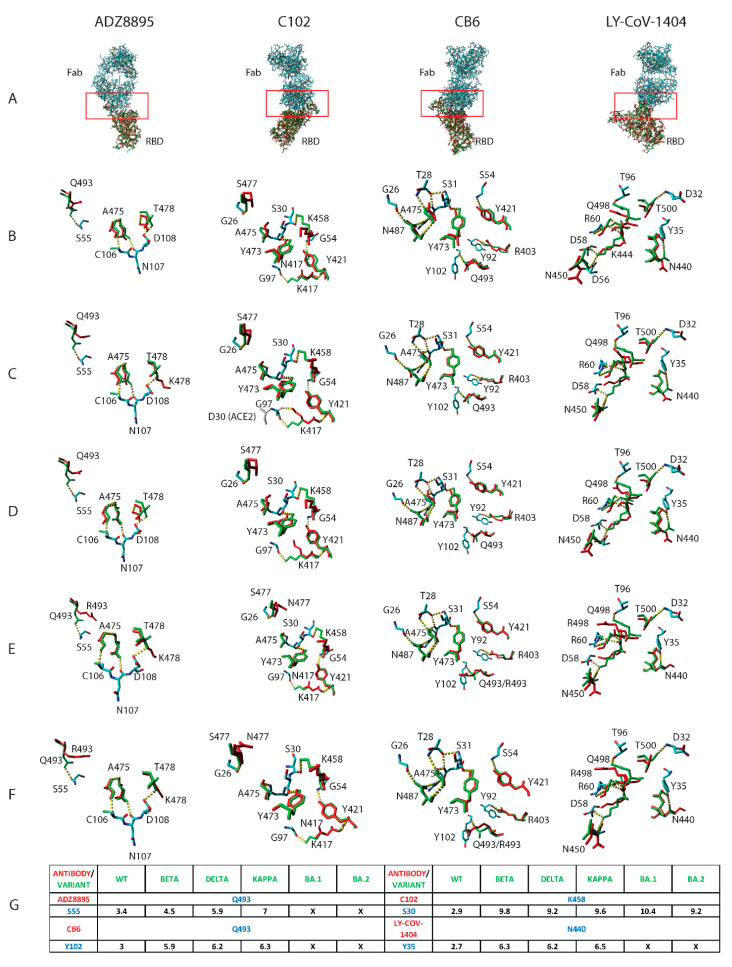
Interaction of ADZ8895, C102, CB6, and LY-CoV-1404 antibodies with SARS-CoV-2 variants of concern (VOC) RBD proteins. Structural alignment of the original RBD protein (green sticks) with the VOC RBD proteins (red sticks) in complex with various monoclonal antibodies (element sticks) (**A**). Comparison of the structural positioning of corresponding epitope amino acids in the original RBD protein (green sticks) with the Beta (**B**); Delta (**C**); Kappa (**D**); BA1 (**E**); and BA2 (**F**) RBD proteins (red sticks). Red boxes and dotted yellow lines represent the paratope-epitope interface and hydrogen bonds, respectively. Distance in Angstroms (black, X represents mutations) between interacting AASCs (blue) on VOC RBD (green) and antibody (red) paratopes (**G**).

**Figure 2 pathogens-11-01437-f002:**
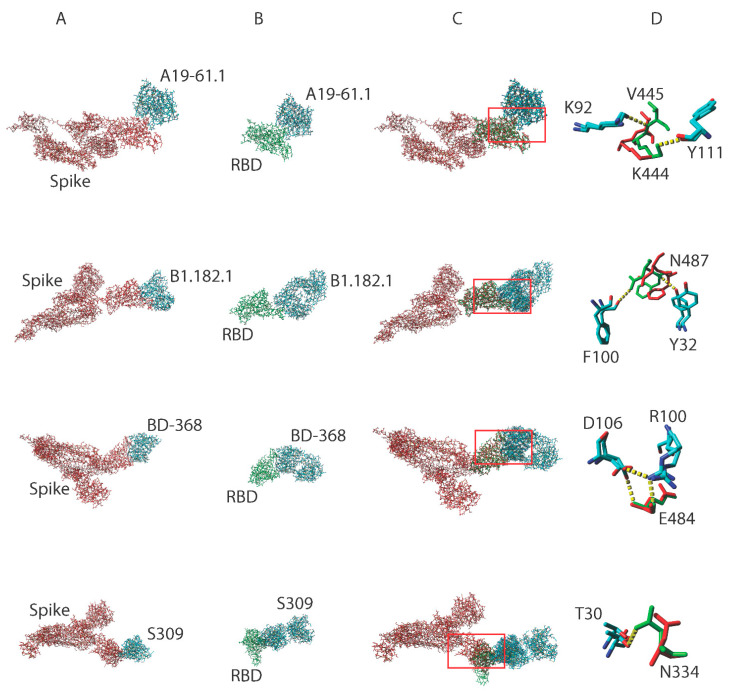
Interaction of A19-61.1, B1.182.1, BD368, and S309 antibodies with SARS-CoV-2 spike and RBD proteins. Antibodies in complex with the spike protein (red sticks) (**A**) and the RBD protein (green sticks) (**B**). Structural alignment of the spike protein against the RBD protein in complex with various antibodies (element sticks) (**C**). Comparison of the structural positioning of AASCs in the spike protein (red sticks) with corresponding AASCs in the RBD protein (green sticks) (**D**). Hydrogen bonds are depicted as dotted yellow lines while red boxes represent the paratope-epitope interface.

**Figure 3 pathogens-11-01437-f003:**
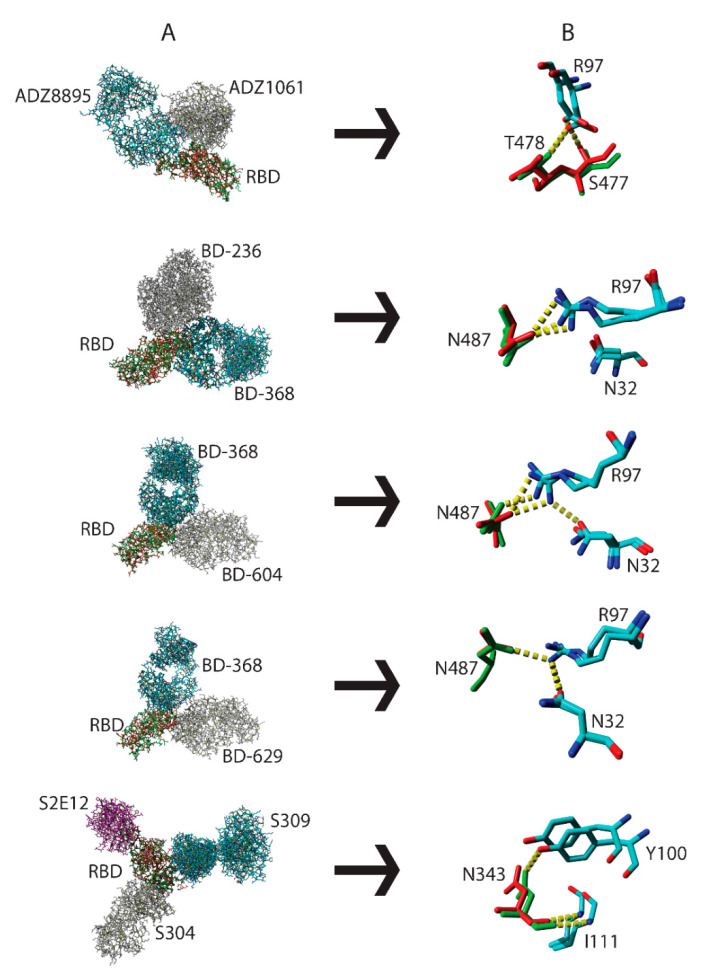
Differential binding of ADZ8895, BD-368, and S309 antibodies to RBD when used individually or as antibody cocktails. Structural alignment of the same SARS-CoV-2 variant RBD protein in complex with individual antibodies or a cocktail of antibodies (**A**). Exposure of selected paratope-epitope AASCs with RBD proteins presented as red (antibody cocktail) and green (individual antibody) sticks while antibodies are depicted as element sticks (**B**). Hydrogen bonds are presented as dotted yellow lines.

**Figure 4 pathogens-11-01437-f004:**
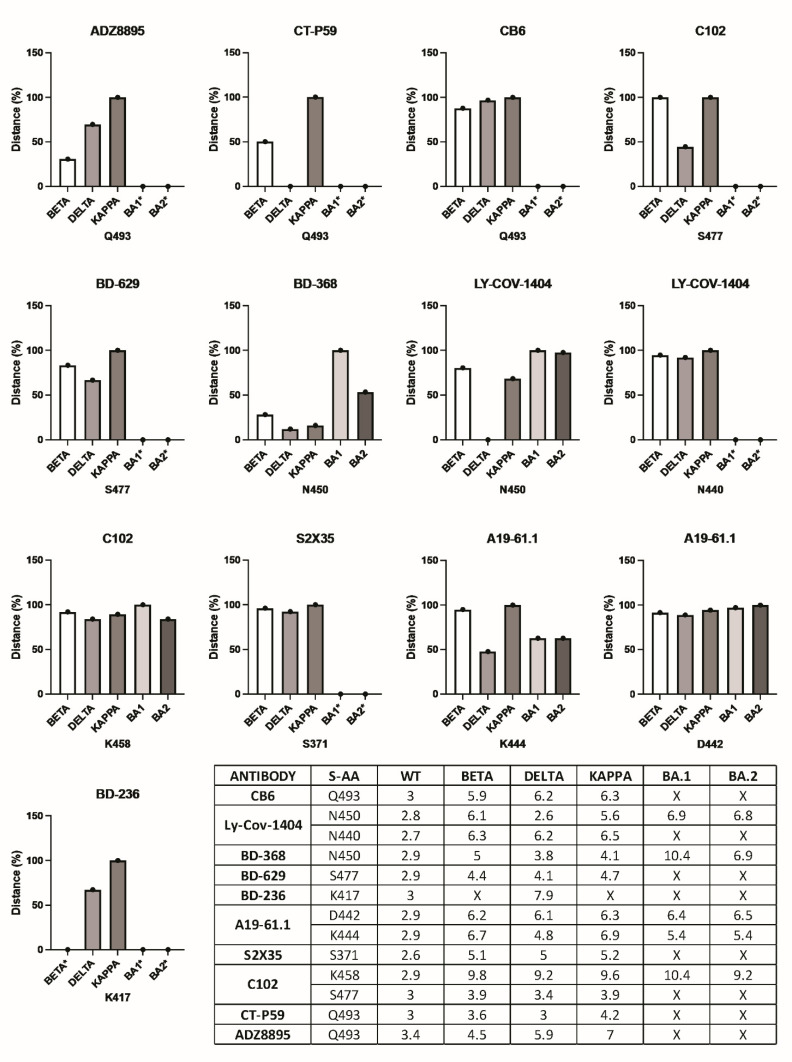
Epitope AASCs in the SARS-CoV-2 spike protein experiencing immune pressure. The actual distances in Angstroms illustrated in the table were converted to percentages and then used to plot histograms. Asterisk (*) in histograms and X in the table represent immune pressure-driven substitutions of epitope amino acids. S-AA—spike amino acids.

**Table 1 pathogens-11-01437-t001:** Characteristics of 3D structures included in this study.

PDB ID	Protein/Antibody Name (Mutations)	Target Protein	Antibody Formulation	Virus/Variant	Expression System	Method
7MMO	LY-CoV1404	RBD	Monoclonal	SARS-CoV-2 wild type	Cricetulus griseus	X-ray diffraction
6WPS	S309	Spike	Monoclonal		Homo sapiens	Electron microscopy
7CHH	BD-368-2		Monoclonal			
7TB8	B1-182.1 and A19-61.1		Cocktail			
7C01	CB6	RBD	Monoclonal			X-ray diffraction
7K8M	C102		Monoclonal			
7L7D	AZD8895		Monoclonal			
7CH4	BD-604		Monoclonal			
7CH5	BD-629		Monoclonal			
7CHB	BD-236		Monoclonal			
7L7E	AZD8895 and AZD1061		Cocktail			
7R6W	S2X35 and S309		Cocktail			
7R6X	S2E12, S309, and S304		Cocktail			
7TBF	B1-182.1 and A19-61.1		Cocktail			
7CHC	BD-629 and BD-368-2		Cocktail			
7CHE	BD-236 and BD-368-2		Cocktail			
7CHF	BD-604 and BD-368-2		Cocktail			
7TN0	ACE2, S304, and S309		Cocktail	SARS-CoV-2 Omicron		
7VX4	ACE2-RBD (K417N, E484K, N501Y)		Monoclonal	SARS-CoV-2 Beta		Electron microscopy
7VX5	ACE2-RBD (L452R and E484Q)			SARS-CoV-2 Kappa		
7WBP	ACE2-RBD (G339D, S371L, S373P, S375F, K417N, N440K, G446S, S477N, T478K, E484A, Q493R, G496S, Q498R, N501Y, Y505H)			SARS-CoV-2 BA.1		X-ray diffraction
7WBQ	ACE2-RBD (L452R, T478K)			SARS-CoV-2 Delta		
7ZF7	ACE2-RBD (G339D, S371L, S373P, S375F, T376A, R408S, K417N, N440K, S477N, T478K, E484A, Q493R, Q498R, N501Y, Y505H)			SARS-CoV-2 BA.2		
2NY7	B12	gp120	Monoclonal	HIV	Cricetulus griseus	
1HZH	B12 IgG		N/A			
5Y2L	AF4H1K1	Hemagglutinin	Monoclonal	Influenza H3N2	Homo sapiens	
5Y2M				Influenza H4N6		
5Y2K			N/A	No ligand		
6J9O	AF4H1K1 scFv				Escherichia coli	
4O5I	F045-092		Monoclonal	A/Victoria/361/2011 (H3N2)	Trichoplusia ni	
4O58				A/Victoria/3/1975 (H3N2)		
5KEL	c2G4 and c13C6	EBOV GP	Cocktail	Ebola	Nicotiana benthamiana	Electron microscopy
5KEN	c4G7 and c13C6					
6DFJ	Z021	Envelope protein DIII	Monoclonal	DENV-1	Homo sapiens	X-ray diffraction
6DFI				Zika		

## Data Availability

Not applicable.

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
