# Peer review of "SARS-CoV-2 Antibody Effectiveness Is Influenced by Non-Epitope Mutation/Binding-Induced Denaturation of the Epitope 3D Architecture"

_pathogens, 2022, doi:10.3390/pathogens11121437_

Round 1

Reviewer 1 Report

SARS-CoV-2 antibody effectiveness is influenced by non-epitope mutation/binding-induced denaturation of the epitope 3D architecture

Moffat M Malisheni, Paul A. MacAry, Albert A. Rizvanov and Mathew Bates

Article titled “SARS-CoV-2 antibody effectiveness is influenced by non-epitope mutation/binding-induced denaturation of the epitope 3D architecture” submitted for publication in “Pathogens” is a well written manuscript focusing on the rationalization of the decrease in the effectiveness of SARS-CoV-2 antibodies caused by the mutation in the spiked protein. Authored have highlighted the role of the non-epitope regions on the binding of the RBD domain of the native and evolving variants of SARS-CoV with therapeutic antibodies and AEC2. Authors have used the bioinformatics tools and available structure to evaluate the interaction of the AASCs for the epitopes in the RBD region with the studies targets. The work highlights the significance of the factors influencing the tertiary structure of the proteins and thus its function. This work shed light into the possible reason behind the loss in the effectiveness of the neutralizing antibodies against evolving variant of SARS-COV-2 virus at the same time their improved transmissibility. I think the work manuscript is very well written with required background, well-described hypothesis and implication of the work. Authors are also aware of some of the limitations of the work described in this study, (e.g. only the hydrogen bond interactions were studied), however the finds are still useful to describe several observations used as case studies.

Similar work focused on rationalize the impact of the distant mutations in the protein on the structure function and stability of the proteins is available in the literature. Authors have done great job in leveraging bioinformatics tool and applying existing theories to shed some insights into the evolving functionality of the SARS-CoV-2 spike protein enabling it to the skip the defence of the neutralizing anitbidies at the same time improves its transmissibility. Authors are however encouraged to do a final proofreading for grammar and typos

Author Response

Thank you very much for your time and positive feedback. We have updated the manuscript. 

Reviewer 2 Report

  • Major comments: 

In the manuscript, Malisheni et al. provided a structural analysis of the PDB structures of SARS-CoV-2 Spike/RBD proteins in complex with either neutralizing antibodies or ACE2. They provided their viewpoints on the structural and mechanistic basis for neutralizing antibody evasion driven by non-epitope amino acid substitutions in the RBD.

The study is helpful in our understanding of the effects of viral non-epitope mutations on the process of immune escape and ACE2 binding. The findings of the study might be helpful in antibody and vaccine developments against SARS-CoV-2.

Major concerns:

The epitope amino acid and non-epitope amino acid in the SARS-CoV-2 RBD region should be listed for clarification. And for epitope amino acid in the SARS-CoV-2 RBD region, the corresponding paratope amino acid in the regions ACE2/antibodies should be listed as well.

It should be noted that the factors like the resolution of the structures and the methods for the structure determination, as well as the expression system of the proteins, might impact the distances of the interacting residues of the epitope-paratope interface. The glycans embedded in the SARS-CoV-2/ACE2/antibodies might also participate in the epitope-paratope interactions.

The study is informative and can be further improved by including the following suggestions listed in specific comments.

  • Specific comments:

1)      Table 1. The title is not appropriate, as some structures are resolved by other methods, like the structure of PDB: 6WPS is resolved by ELECTRON MICROSCOPY. Besides, the methods of the structure determination for each PDB included in the study should be listed in Table 1 as well. Different methods, like crystallography, NMR, or Cryogenic electron microscopy, might generate slightly different conformations even for the same paratope-epitope interface. For the PDB of 7VX4, 7VX5, 7WBP, 7WBQ, and 7ZF7, the mutations inside these SARS-CoV-2 RBD variants should be listed in the Table legend for clarification. 7TN0 was Omicron RBD, not wildtype RBD. Please double-check all PDBs in Table 1.

2)      Line 138, what does “N” in the equation stands for?

3)      Figure 1. Figure 1B, Figure 1C, Figure 1D, Figure 1E, and Figure 1F, the amino acid names/positions that participated in the paratope-epitope interface interactions should be marked in the corresponding positions in Figure 1 for clarification.

4)      Line 154-158, and Figure 1. Please provide a detailed analysis of the results in Figure 1 to support the conclusion.

5)      Figure S1. Figure S1B, Figure S1C, Figure S1D, and Figure S1E, the amino acid names/positions that participated in the paratope-epitope interface interactions should be marked in the corresponding positions in Figure S1 for clarification.

6)      Figure S2. The pictures in the left panel should be labeled as “A,” and the table in the right panel should be labeled as “B.”

7)      For Figure S2B, the BA1 variant also bears the mutations of K417N, G446S, and G496S; the BA2 variant also contains the mutations of K417N and G446S. BA2 variant does not contain the mutation of G496S, though.

8)      For Figure S2B, the letter “X” represents the substituted amino acids? It seems like WT, BETA, KAPPA, and DELTA RBD don’t bear Q498 mutations.

9)      Lines 173-174, it seems like ACE2 Q42 also interacts with G446 on the original RBD.

10)  Line 174-175, ACE2 K353 interacts with Q498 on the original RBD? Actually, it is ACE2 Q42 that interacts with Q498 on the original RBD.

11)  Line 190, type error “AARS.”

12)  Line 192-194 and Figure 2D. Please provide a detailed analysis of the results in Figure 2D and illustrate how the differences of spike protein with RBD influenced the formation of hydrogen bonds in the paratope epitope interface. Besides, in Figure 2D, the amino acid names/positions that participated in the paratope-epitope interface interactions should be marked in the corresponding positions for clarification.

13)  Figure S3, Figure S3B, Figure S3C, Figure S3D, and Figure S3E, the amino acid names/positions that participated in the paratope-epitope interface interactions should be marked in the corresponding positions for clarification. Figure S3F, Surface (gray) representation of HIV gp120?

14)  Line 201 and Figure S3F, please mark the position of B12 R28 in Figure S3F.

15)  Line 209 and Line 217, SARS-CoV-2 variant RBD protein here? Please confirm the RBD information here from Table 1.

16)  Line 220-222, Please provide a detailed description here, and illustrate the three distinctive hydrogen bond interactions of BD-368.

17)  Line 244 and Figure S4(C13C6), there is no C4G7 antibody showed in Figure S4(C13C6). In addition, what do red and green sticks stand for in Figure S4(C13C6)? In Figure S4B(C13C6), the amino acid names/positions that participated in the paratope-epitope interface interactions should be marked in the corresponding positions for clarification.

18)  Line 244-245. Please provide a detailed description here.

19)  Line 230-232, well, N477(Omicron RBD here) is able to interact with S19(ACE2) individually in other studies. In addition, what do red and green sticks stand for in Figure S4(ACE2)?

20)  Figure 4, table panel, it seems like the Beta variant also bears K417N mutation.

21)  Figure 4, Figure panel, all histograms used the KAPPA variant as a standard comparison (100%)?

22)  Figure S5, Figure S5B, Figure S5C, Figure S5D, Figure S5E, and Figure S5F, even though SARS-CoV-2 wildtype and BA.2 RBD proteins both used the element sticks here, it would be better to use different colors to differentiate these TWO RBD proteins for better illustration. In addition, the amino acid names/positions of the paratope side should be marked in the corresponding positions of Figure S5B, Figure S5C, Figure S5D, Figure S5E, and Figure S5F for clarification.

23)  Line 279-281, In some studies/structures, Q24 in wildtype RBD already interacted with N487(ACE2) through hydrogen bonds.

24)  Figure S6H. Even though the original and Kappa epitope AASCs are presented as element sticks here, it would be better to use different colors to differentiate these TWO epitope AASCs for better illustration.

25)  Figure S6 (I – L). It would be better to use different colors to differentiate SARS-CoV-2 (N487) from ACE2 (Q24).

26)  Line 340-345, according to the literature, COVID-19 booster vaccines containing spike proteins from the original strain still work even for Omicron. The updated bivalent COVID-19 vaccines also contain spike proteins from the original strain, along with BA.4/5 SARS-CoV-2 variants. The literature should be discussed further here.

Author Response

Thank you for your time and suggestions meant to improve our manuscript. Please find our responses to your comments below.

Specific comments:

1)      Table 1. The title is not appropriate, as some structures are resolved by other methods, like the structure of PDB: 6WPS is resolved by ELECTRON MICROSCOPY. Besides, the methods of the structure determination for each PDB included in the study should be listed in Table 1 as well. Different methods, like crystallography, NMR, or Cryogenic electron microscopy, might generate slightly different conformations even for the same paratope-epitope interface. For the PDB of 7VX4, 7VX5, 7WBP, 7WBQ, and 7ZF7, the mutations inside these SARS-CoV-2 RBD variants should be listed in the Table legend for clarification. 7TN0 was Omicron RBD, not wildtype RBD. Please double-check all PDBs in Table 1.

Authors’ responses

Changes were made as advised. Please see the revised manuscript and supplementary material.

2)      Line 138, what does “N” in the equation stands for?

Authors’ responses

N was defined in the next sentence. Line 148.

3)      Figure 1. Figure 1B, Figure 1C, Figure 1D, Figure 1E, and Figure 1F, the amino acid names/positions that participated in the paratope-epitope interface interactions should be marked in the corresponding positions in Figure 1 for clarification.

Authors’ responses

Changes were made as advised. Please see the revised manuscript.

4)      Line 154-158, and Figure 1. Please provide a detailed analysis of the results in Figure 1 to support the conclusion.

Authors’ responses

Changes were made as advised. Please see the revised manuscript.

5)      Figure S1. Figure S1B, Figure S1C, Figure S1D, and Figure S1E, the amino acid names/positions that participated in the paratope-epitope interface interactions should be marked in the corresponding positions in Figure S1 for clarification.

Authors’ responses

Changes were made as advised. Please see the revised supplementary material.

6)      Figure S2. The pictures in the left panel should be labeled as “A,” and the table in the right panel should be labeled as “B.”

Authors’ responses

Changes were made as advised. Please see the revised supplementary material.

7)      For Figure S2B, the BA1 variant also bears the mutations of K417N, G446S, and G496S; the BA2 variant also contains the mutations of K417N and G446S. BA2 variant does not contain the mutation of G496S, though.

Authors’ responses

Changes were made as advised. Please see the revised supplementary material.

8)      For Figure S2B, the letter “X” represents the substituted amino acids? It seems like WT, BETA, KAPPA, and DELTA RBD don’t bear Q498 mutations.

Authors’ responses

Changes were made as advised. Please see the revised manuscript and supplementary material.

9)      Lines 173-174, it seems like ACE2 Q42 also interacts with G446 on the original RBD.

Authors’ responses

ACE2 Q24 did not interact with G446 because of the distance. The numbers were included to show that the distances were > 3.5, above which hydrogen bonds cannot form.

10)  Line 174-175, ACE2 K353 interacts with Q498 on the original RBD? Actually, it is ACE2 Q42 that interacts with Q498 on the original RBD.

Authors’ responses

This has been corrected accordingly.

11)  Line 190, type error “AARS.”

Authors’ responses

This has been corrected accordingly.

12)  Line 192-194 and Figure 2D. Please provide a detailed analysis of the results in Figure 2D and illustrate how the differences of spike protein with RBD influenced the formation of hydrogen bonds in the paratope epitope interface. Besides, in Figure 2D, the amino acid names/positions that participated in the paratope-epitope interface interactions should be marked in the corresponding positions for clarification.

Authors’ responses

Changes were made as advised. Please see the revised manuscript.

13)  Figure S3, Figure S3B, Figure S3C, Figure S3D, and Figure S3E, the amino acid names/positions that participated in the paratope-epitope interface interactions should be marked in the corresponding positions for clarification. Figure S3F, Surface (gray) representation of HIV gp120?

Authors’ responses

Changes were made as advised. Please see the revised supplementary material.

14)  Line 201 and Figure S3F, please mark the position of B12 R28 in Figure S3F.

Authors’ responses

Changes were made as advised. Please see the revised supplementary material.

15)  Line 209 and Line 217, SARS-CoV-2 variant RBD protein here? Please confirm the RBD information here from Table 1.

Authors’ responses

This information was confirmed.

16)  Line 220-222, Please provide a detailed description here, and illustrate the three distinctive hydrogen bond interactions of BD-368.

Authors’ responses

Changes were made as advised. Please see the revised manuscript.

17)  Line 244 and Figure S4(C13C6), there is no C4G7 antibody showed in Figure S4(C13C6). In addition, what do red and green sticks stand for in Figure S4(C13C6)? In Figure S4B(C13C6), the amino acid names/positions that participated in the paratope-epitope interface interactions should be marked in the corresponding positions for clarification.

Authors’ responses

Changes were made as advised and the sticks were labeled accordingly. Please see the revised supplementary material.

18)  Line 244-245. Please provide a detailed description here.

Authors’ responses

Changes were made as advised. Please see the revised manuscript.

19)  Line 230-232, well, N477(Omicron RBD here) is able to interact with S19(ACE2) individually in other studies. In addition, what do red and green sticks stand for in Figure S4(ACE2)?

Authors’ responses

That is correct. As you mentioned earlier that structural differences are possible when different methods are used. This was highlighted in the study limitations.

20)  Figure 4, table panel, it seems like the Beta variant also bears K417N mutation.

Authors’ responses

The table was revised accordingly. Please see the revised manuscript and supplementary material.

21)  Figure 4, Figure panel, all histograms used the KAPPA variant as a standard comparison (100%)?

Authors’ responses

The comparator was the wild-type protein.

22)  Figure S5, Figure S5B, Figure S5C, Figure S5D, Figure S5E, and Figure S5F, even though SARS-CoV-2 wildtype and BA.2 RBD proteins both used the element sticks here, it would be better to use different colors to differentiate these TWO RBD proteins for better illustration. In addition, the amino acid names/positions of the paratope side should be marked in the corresponding positions of Figure S5B, Figure S5C, Figure S5D, Figure S5E, and Figure S5F for clarification.

Authors’ responses

Changes were made as advised. Please see the revised supplementary material.

23)  Line 279-281, In some studies/structures, Q24 in wildtype RBD already interacted with N487(ACE2) through hydrogen bonds.

Authors’ responses

That is correct. As you mentioned earlier that structural differences are possible when different methods are used. This was highlighted in the study limitations.

24)  Figure S6H. Even though the original and Kappa epitope AASCs are presented as element sticks here, it would be better to use different colors to differentiate these TWO epitope AASCs for better illustration.

Authors’ responses

Changes were made as advised. Please see the revised supplementary material.

25)  Figure S6 (I – L). It would be better to use different colors to differentiate SARS-CoV-2 (N487) from ACE2 (Q24).

Authors’ responses

Changes were made as advised. Please see the revised supplementary material.

26)  Line 340-345, according to the literature, COVID-19 booster vaccines containing spike proteins from the original strain still work even for Omicron. The updated bivalent COVID-19 vaccines also contain spike proteins from the original strain, along with BA.4/5 SARS-CoV-2 variants. The literature should be discussed further here.

Authors’ responses

Changes were made as advised. Please see the revised manuscript.

Round 2

Reviewer 2 Report

I think the authors have sufficiently addressed most of my concerns.

One more suggestion:

Line 85 “2.1. Acquisition of crystal structures” and Line 112 “2.2. Analysis of crystal structures”, please modify the titles here as not all structures are resolved by crystallography.